

# Detecting security attacks in cyber-physical systems: a comparison of Mule and WSO2 intelligent IoT architectures

José Roldán-Gómez[1], Juan Boubeta-Puig[2], Gabriela Pachacama-Castillo[3], Guadalupe Ortiz[2] and Jose Luis Martínez[1]

[1] Research Institute of Informatics (i3a), Universidad de Castilla La Mancha, Albacete, Spain
[2] Department of Computer Science and Engineering, University of Cadiz, Cadiz, Spain
[3] School of Engineering, University of Cadiz, Cadiz, Spain

Corresponding author
José Roldán-Gómez,
jose.roldan@uclm.es

## ABSTRACT

The Internet of Things (IoT) paradigm keeps growing, and many different IoT devices, such as smartphones and smart appliances, are extensively used in smart industries and smart cities. The benefits of this paradigm are obvious, but these IoT environments have brought with them new challenges, such as detecting and combating cybersecurity attacks against cyber-physical systems. This paper addresses the real-time detection of security attacks in these IoT systems through the combined used of Machine Learning (ML) techniques and Complex Event Processing (CEP). In this regard, in the past we proposed an intelligent architecture that integrates ML with CEP, and which permits the definition of event patterns for the real-time detection of not only specific IoT security attacks, but also novel attacks that have not previously been defined. Our current concern, and the main objective of this paper, is to ensure that the architecture is not necessarily linked to specific vendor technologies and that it can be implemented with other vendor technologies while maintaining its correct functionality. We also set out to evaluate and compare the performance and benefits of alternative implementations. This is why the proposed architecture has been implemented by using technologies from different vendors: firstly, the Mule Enterprise Service Bus (ESB) together with the Esper CEP engine; and secondly, the WSO2 ESB with the Siddhi CEP engine. Both implementations have been tested in terms of performance and stress, and they are compared and discussed in this paper. The results obtained demonstrate that both implementations are suitable and effective, but also that there are notable differences between them: the Mule-based architecture is faster when the architecture makes use of two message broker topics and compares different types of events, while the WSO2-based one is faster when there is a single topic and one event type, and the system has a heavy workload.

# INTRODUCTION

Over the past few years, expectations regarding the use of IoT devices have risen significantly. According to data published by the IoT Analytics company, since 2015 there has been a significant increase in the use of IoT devices, with 7,000 million of them being

registered in 2018, and this figure is estimated to reach 21,500 million in 2025 (*Lueth, 2018*). With this increase in the use of such devices, new security challenges also arise, such as ensuring the security of IoT devices (*Bertino et al., 2016*). Although there are quite a number of works in the literature addressing this problem, further research and implementation is still needed within the realm of the Internet of Things. An example of this is the attack in 2016 in which cybercriminals exploited the vulnerabilities of thousands of IoT devices to convert them into Domain Name System (DNS) request generators and carry out a Distributed Denial of Service (DDoS) attack, causing an Internet service disruption that affected several companies such as Amazon, PayPal, Netflix, Spotify and Twitter (*Moss, 2016*). It is also worth mentioning that the analysis published by the Gartner company indicated that in 2020 more than 25% of the attacks identified in companies would involve IoT devices (*Moore, 2018*). Several studies show the magnitude of the problem, revealing that, in just the first half of 2019, a hundred million attacks were carried out against smart devices, a figure seven times higher than the number detected in 2018. The Mirai malware was responsible for 39% of them (*Demeter, Preuss & Shmelev, 2019*). In 2020 and 2021 this problem has worsened; the most common threat remains Mirai, but new variants have also been created (*Gutnikov et al., 2021*; *Kaspersky, 2021*).

Considering all of the above, it is clear that there are currently significant security problems in IoT devices; if these problems are not addressed, it is certain that they will be even more damaging in the future. Therefore, it is imperative to examine new ways of identifying attacks on IoT devices in a timely and efficient manner, and to enable notification and alarm submission in critical attacks. In other words, it is essential to propose an Intrusion Detection System (IDS) for IoT devices. Such a system must be able to receive, analyze and process a large number of records in real time. Also, it must immediately notify security experts of attacks in progress in order to give them more reaction time to mitigate the attacks. However, the ability of traditional rule-based IDSs to detect security attacks in the IoT domain is limited, as they cannot detect novel attacks. Since current malware is not static, it is highly desirable to have the ability to detect previously-unknown attacks.

With the aim of addressing this challenge, in the past we proposed a software architecture that integrates Complex Event Processing (CEP) and Machine Learning (ML), and has the ability to detect, and provide notification of, security attacks on IoT devices in real time (*Roldán et al., 2020*). This architecture permits the detection of not only static but also dynamic security attacks in the IoT thanks to the use of both CEP technology (*Luckham, 2012*; *Boubeta-Puig, Ortiz & Medina-Bulo (2015)*) and ML techniques (*Buczak & Guven, 2016*).

Once we had proved the viability of building this architecture by combining CEP and ML, we observed a number of potential limitations that should be studied; in particular, we were concerned with the fact that the architecture is necessarily linked to the technologies of specific vendors, and also that other alternative implementations may not achieve the desired performance in this field of application. The architecture we proposed consists of the integration of ML with a CEP engine, and the ESB of two specific

vendors, namely Esper (*EsperTech, 2021*) and MuleSoft (*MuleSoft, 2021b*). We thus thought it might be advisable to be able to implement this architecture on other platforms; for example, on the well-known WSO2 suite (*WSO2, 2021c*).

This gave rise to our first research question (RQ1): can a real-time data stream processing architecture be implemented with the WSO2 ESB (*WSO2, 2021d*) together with the WSO2 Siddhi CEP engine (*WSO2, 2021b*) and be integrated with ML techniques? Assuming that it is feasible to implement the architecture with the CEP engine and the ESB of other vendors, in particular with those offered by WSO2, we are necessarily concerned about what impact this may have on the performance of the system, given that, as we have explained above, a real-time response is required to stop security attacks on the IoT.

This leads us to our second research question (RQ2): can a streaming data processing architecture based on the integration of ML techniques with the WSO2 CEP engine and ESB achieve or improve upon the performance of the previously proposed architecture (*Roldán et al., 2020*)? In addition, we consider the possibility that various implementations of the integration architecture of CEP, ESB and ML may present a more or less advantageous performance depending on the type of attack to be detected, that is, the type of pattern necessary for each attack. Likewise, there may be variations in how these systems support situations of stress.

This inevitably leads us to the third research question (RQ3): what kind of event patterns are processed faster with WSO2/Siddhi and which ones with Mule/Esper, and which of the two architectures is more suitable for supporting high-stress situations?

Once all this analysis has been carried out, we undoubtedly arrive at the question in which the domain experts are most interested (RQ4): which of these architecture implementations is the best to be deployed in an IoT security attack detection environment? To be able to answer these research questions requires the implementation of the architecture analogous to the one presented in *Roldán et al. (2020)* and replacing the technologies by the ones in the WSO2 suite. It also requires the implementation of a realistic security attack environment in an IoT network by carrying out various attacks against the TCP, UDP and MQTT protocols, as well as analyzing the response of the architectures in terms of performance and stress tests.

Therefore, the main aim of this paper is twofold: firstly, we aim to demonstrate that our intelligent architecture, which integrates CEP and ML in order to detect IoT security attacks in real time, can be implemented with different integration platforms such as Mule and WSO2, different CEP engines such as Esper and Siddhi and different ML algorithms such as linear regression (*Montgomery, Peck & Vining, 2021*). Secondly we aim to provide a comprehensive analysis of the performance and benefits of the architecture depending on the different vendor technologies used for its implementation; in particular, a comparison of the architecture implementation with Mule and Esper *versus* WSO2 and Siddhi is included. In this way, we provide a comparative analysis that can be very useful for the developer when choosing between one technology and another for the implementation of the architecture, depending on the requirements of the specific application domain and case study.

In addition to the research questions and the objectives to be achieved, in this work we rely on a series of assumptions that can be extracted from different works, These are:

- CEP works successfully in IoT environments. There are different works in which CEP architectures are successfully deployed in IoT environments (*Roldán et al., 2020*; *Corral-Plaza et al., 2020*).

- CEP engines and ESBs from different vendors can be integrated with our architecture to detect cybersecurity threats in real time: this architecture has already been deployed with Mule (*Roldán et al., 2020*) and there are works describing how to deploy WSO2 in an IoT environment (*Fremantle, 2015*).

The rest of the paper is organized as follows. The *Background* section describes the background to the paradigms and technologies used in this work. The *Related work* section describes the most relevant works in the literature, and the *Architecture for IoT security* section presents the architecture we propose for detecting attacks on IoT devices and how the implementation with the WSO2 suite differs from that of Esper CEP and Mule ESB. The *Comparing architecture performance and stress* section explains the comparison of the performance and stress tests conducted for these architectures, which have been implemented with Esper/Mule and WSO2. Then, the *Results* section presents the experiments and results obtained, the *Discussion* section discuss and answer the four research questions. Finally, the *Conclusions and future work* section contains our conclusions and some lines for future research.

## BACKGROUND

This section describes the background to security in the IoT, ML, SOA 2.0 and CEP.

### Security in the Internet of things

The IoT and cyber-physical system devices are increasingly present in our lives. The features offered by these devices are very attractive and they can be used for many different purposes, among which, we can highlight domotics, the automation and control of production processes, video surveillance and security, and medicine and health care. The various uses that have been given to these devices and the ability to access them *via* the Internet have attracted the interest of hackers. Unfortunately, the approach followed by developers in the design of security measures for IoT devices has not been as successful as their growth, and this is made evident by the number of cyber-attacks detected in the first half of 2019, which surpassed a hundred million, which is seven times higher than the previous year (*Demeter, Preuss & Shmelev, 2019*). The vector used by attackers in those attacks was mainly brute force, taking advantage of the weak default configuration of the devices and gaining access to them with the default credentials (*Demeter, Preuss & Shmelev, 2019*). These attacks took advantage of the vulnerabilities of the IoT devices to infect them with malicious code and then manipulate them to achieve their goal. The idea behind that malware focused on the creation of bots to be marketed for the carrying out of Denial of Service (DoS) attacks. One of the most widely-spread (and also the first)

pieces of malware specially designed for these devices was called Mirai, which is a botnet that inserts malicious code into IoT devices so that they initiate a DoS attack against a certain target. This caused shock and aroused the interest of hackers in these devices.

Another weakness of IoT and cyber-physical system devices is the use of unsafe network services and protocols, due mainly to these devices having several constraints, such as a small memory and a limited battery, which prevent developers from using a usual security setup. These vulnerabilities have been exploited to carry out several attacks that could have been prevented if the necessary measures had been taken. A lack of security in the storage and transfer of data that allows the observation and analysis of the information transmitted by these devices is another critical weakness in the security of IoT devices. In this regard, Message Queuing Telemetry Transport (MQTT) is a very common protocol in the IoT (*OASIS, 2019*). MQTT is a binary protocol that reduces the overhead compared with other application layer protocols. It is a publish/subscribe-based protocol in which a server (there can be more than one), known as the message broker, manages the flow of information, which is organized as a hierarchy of topics. Each client can be a subscriber and a publisher simultaneously. This protocol is similar to MQTT-SN and has several weaknesses, such as allowing the sending of many MQTT packets of a massive size, which overloads the broker. This attack causes a DoS in the MQTT network. Furthermore, an MQTT subscription fuzzing attack could gain information about the available topics because nodes are not authenticated and the information is not ciphered. Moreover, an MQTT disc-wave attack can exploit a failing in several implementations of the MQTT protocol. The specification of MQTT establishes that each client has a unique ID, so if a new client tries to register this ID again, the broker should reject it. However, many implementations allow a new client to connect with a registered ID, causing the existing client with that ID to be ejected from the previously-created connection.

Finally, a very common attack that can appear in an IoT-based network is scanning. Attackers can perform this procedure to discover devices and open ports in the network. By extending the scanning, attackers can cause a DoS in the network by sending large numbers of reconnaissance packets and congesting the network. The attack generates a large volume of traffic to try to saturate the network and so prevent users from accessing the system. The attack can also take advantage of flaws in the code of an application or part of the open-source code that uses the application. Two of the most common attacks of this type are TCP and UDP flood attacks (*Warburton, 2021*). When the connection is established through the TCP protocol, the client and the server exchange flags to initiate, close or restart the connection, or indicate that the request is urgent; the attacker sends several SYN flags asking to initiate a connection with the server, which is blocked when there are too many ACK requests waiting and the server runs out of resources to serve legitimate clients. A UDP port scan attack consists of sending a UDP packet to multiple ports on the same destination system, then analyzing the response and determining service and host availability. The attacker can determine whether the port is open, closed or filtered through a firewall or packet filter.

## Machine learning

Machine Learning (ML) can be described as a set of techniques, technologies, algorithms and methodologies used to predict, cluster and classify entities, which can be events, objects, or anything else that can be described with attributes, also known as features, and entity behaviors. Broadly speaking, the best way to obtain these predictions is to model the behavior and attributes of these entities. There are many different algorithms to model these entities using functions which are plotted with these algorithms, and datasets of entities. The behavior, features and context of each entity are different. Therefore, the best algorithm does not exist, as each entity type has its correct algorithm or algorithms, if they even exist. For this reason, it is necessary to analyze these entities and their contexts, preprocess the datasets to allow them to be managed by these algorithms, and perform a feature selection (if it is necessary) to discover the most descriptive set of features. Sometimes, once the feature selection has been made, we can easily obtain the distribution of the entities, which is very useful for choosing the algorithm in a more precise way.

There are different types of machine learning techniques and algorithms, which can be classified as follows:

- Supervised learning. In this approach, the model is trained with labelled entities, *i.e.* the model knows the type of each entity in the training dataset. Also, it is possible to find regression techniques that aim to predict a numeric value.
- Unsupervised learning. This set of techniques does not require labelled entities, so the model learns how to group or classify them with similarity measures.
- Reinforcement learning. This kind of ML uses a prize/penalty approach. When our model performs a correct action, we can provide it with good feedback. When it fails, then it receives a penalty.

In this paper, we have used linear regression (*Montgomery, Peck & Vining, 2021*) because our dataset has a linear distribution. We would like to highlight that our approach can be adapted to other mathematical models, if needed.

## Event-driven service-oriented architectures

Service-Oriented Architecture (SOA) is a paradigm for the design and implementation of loosely-coupled distributed system architectures whose implementation is fundamentally based on services. SOA services offer a well-defined interface in accordance with standards and facilitate communications between the service provider and the consumer in a decoupled way by using standard protocols. Thus, these architectures provide easy interoperability between third party systems in a flexible way, and therefore facilitate system maintenance and evolution when changes are required (*Papazoglou, 2012*).

ED-SOA, or SOA 2.0, has evolved from the traditional SOA. The distinguishing feature of SOA 2.0 is that it facilitates communication between users, applications and services through events, instead of using remote procedure calls (*Luckham, 2012*). With the growth of service components and processes, and the inclusion of events in event-driven

service-oriented applications, a new infrastructure is required to support the decoupled communications and to maintain applications flexibly. These requirements are fulfilled by an ESB, which permits interoperability among several communication protocols and heterogeneous data sources and targets (*Papazoglou, 2012*). In this way, an ESB provides and supports interoperability among diverse applications and components through standard interfaces and messaging protocols, also reinforcing the reliability of the communication as well as ensuring their scalability. There are several ESBs available, and in this paper we have selected two well-known ones for their evaluation, namely Mule and WSO2.

The Anypoint platform offers support for the design, implementation and management of APIs and integration (*MuleSoft, 2021a*). It includes Mule (*MuleSoft, 2021b*), an integration and ESB platform that provides assistance to developers in interconnecting applications, and provides support for various transport protocols, as well as for the transformation of different data formats. It delivers message routing as well as IoT and cloud integration. In addition, it provides a graphical interface for the development of business-to-business integration applications.

WSO2 is an open-source decentralized approach which provides support for building decoupled digital products that are ready to market, with a main focus on APIs and microservices, and a wide range of complementary products and solutions (*WSO2, 2021c*). WSO2 offers WSO2 Enterprise Integrator, an integration platform which consists of a centralized integration ESB with capabilities for data, process and business-to-business integration. WSO2 ESB (*WSO2, 2021d*) provides support for multiple transport protocols, data formats and flow integration, as well as IoT and cloud service integration. The product also includes an analysis system for comprehensive monitoring.

As we can see, both ESBs provide similar features and can be used in conjunction with their integration platform with many plugins and solutions for further functionalities, such as stream and event processing.

## Complex event processing

Despite all the advantages of SOA 2.0 mentioned in the previous subsection, this type of architecture requires the use of an additional technology that makes it possible to analyze and correlate the vast amounts of data that are present in the field of the IoT in real time. CEP (*Luckham, 2012*) fulfills this functionality appropriately as it is a technology that allows the analysis and correlation of heterogeneous data streams in real time in order to detect situations of interest in the domain in question. In particular, the software that is capable of analyzing the data in real time is known as the CEP engine. In order to detect situations of interest, a series of event patterns are defined in the CEP engine (*Valero et al., 2021*). These patterns represent the conditions that allow us to detect that such a situation has occurred. These rules are applied to the engine's incoming data, which are known as simple events, while the situations of interest detected by the pattern are named complex events. Thus, with CEP we can improve and speed up the decision-making process (*Boubeta-Puig, Ortiz & Medina-Bulo, 2015*; *Benito-Parejo, Merayo & Núñez (2020)*; *Corral-Plaza et al., 2021*).

There are several CEP engines available, and in this paper we have selected two well-known ones to be evaluated, namely Esper and WSO2 CEP.

Esper (*EsperTech, 2021*) is an open-source Java-based software engine for CEP, which can quickly process and analyze large volumes of incoming IoT data. Esper comes with the Esper Event Processing Language (EPL), which extends the SQL standard and permits the precise definition of the complex event patterns to be detected. The Esper compiler compiles EPL into byte code in a JAR file for its deployment, and at runtime this byte code is loaded and executed. Esper performs real-time streaming data processing, using parallelization and multithreading when necessary, and it is highly scalable. In addition, it provides the option of implementing distributed stream processing over several machines as well as horizontal scalability, should it be necessary. According to its documentation, Esper 8.1.0 can process around 7.1 million events per second (*EsperTech, 2019*).

WSO2 CEP is provided within the WSO2 Stream Processor. WSO2 CEP is an open-source CEP engine that facilitates the detection and correlation of events in real time, as well as the notification of alerts, counting in addition with the support of enriched dashboard tools for monitoring. It can be deployed in standalone or distributed modes, and is highly scalable. It uses a streaming processing engine with memory optimization, being able to find patterns of events in real time in milliseconds. According to its specification, a single WSO2 CEP node can handle more than 100 K events per second on a regular 4-core machine with 4 GB of RAM and several million events within the JVM (*WSO2, 2021c*). The cornerstone of the WSO2 CEP is Siddhi (*WSO2, 2021b*). It uses a language similar to SQL that allows complex queries involving time windows, as well as pattern and sequence detection. In addition, CEP queries can be changed at runtime through the use of templates.

ESB has been used in our system as a tool for transport and information management. This use is quite simple to implement but if the parameters are not specified properly, it could cause problems.

## RELATED WORK

There is an interesting comparison between Mule, WSO2 and Talend conducted by *Górski & Pietrasik (2017)*. Note that Talend is beyond the scope of this work. The authors implemented seven different use cases and tested them with 5, 20 and 50 users simultaneously. Moreover, their work provides measurements of throughput, standard deviation and CPU usage for each experiment, and their results are closely aligned with ours, *i.e.*, WSO2 is always faster except when the output message is enormous (221,000 bytes of output message in this case). This is not a problem for our proposal because an IDS does not need big output messages. Moreover, WSO2 obtains a better throughput, whereas the CPU usage is similar in both cases. On the other hand, Mule provides a lower standard deviation, *i.e.*, Mule is more constant than WSO2 when processing different types of events.

Bamhdi's work (*Bamhdi, 2021*) is also interesting. In contrast to our work, his paper does not show an active performance comparison between WSO2 and Mule, but instead

provides a feature comparison between four ESB platforms (WSO2 and Mule are included among them). Although Bamhdi's work is focused on comparing open source platforms against proprietary ones, it allows us to compare specific features of Mule and WSO2. This comparison, which analyzes 15 features, shows that WSO2 supports the 15 listed capabilities, whereas Mule supports. The only feature which Mule cannot provide is web migration from 5.0 to 6.0; note that WSO2 is the only one that satisfies this feature.

*Dayarathna & Perera (2018)* compare WSO2 with other ESBs, but Mule is not considered in their work, which provides a brief feature comparison between the Esper (basic version) and Siddhi CEP engines. According to the authors, each language provided by a CEP engine has its pros and cons. On the one hand, Esper (basic version) provides nested queries and debugging support, while Siddhi registered a higher performance than Esper: 8.55 million events/second *versus* 500,000 events/second.

Another work which is focused on CEP engines is that of *Giatrakos et al. (2020)*. It does not directly compare WSO2 against Mule or Siddhi against Esper, but instead describes different CEP paradigms. In particular, it explains different selection policies, consumption policies and windows. Moreover, the paper describes the scalability and parallelization of several CEP engines. Although it is quite different from our work, it can be useful in order to understand our work and learn about other CEP engines.

*Freire, Frantz & Roos-Frantz (2019)* adopt a different approach in which they do not conduct performance experiments directly, but experts enumerate the features of different ESBs. These features are grouped into three dimensions: message processing, hotspot detection and fairness execution. Additionally, Freire et al.'s work defines two types of features: subjective and objective. The authors assign values for each feature, which allows them to obtain a score for each ESB. According to their paper, Mule should be faster than WSO2, but the problem is that this is not demonstrated through experiments. This approach is useful because it allows the measuring of different ESB platforms without implementing experiments; however, it would have been more useful if they had carried out experiments to support their results.

Our paper provides a real performance comparison between Mule and WSO2, following a similar methodology to the one proposed in the papers mentioned, *i.e.*, executing and deploying the proposed platforms under equal conditions and measuring events in relation to time. More specifically, we have analyzed different pattern types, namely time-window-based patterns and prediction patterns. The latter are a novelty with respect to other works, as each network event is compared with a prediction event, and this acts as an anomaly detector.

## ARCHITECTURE FOR IOT SECURITY

This section describes our proposed SOA 2.0 architecture, which integrates CEP and ML paradigms in order to detect attacks on IoT devices. Then, two implementations of this architecture, one using Mule and the other WSO2, are presented with the aim of comparing them under the same conditions in order to find the strengths and benefits of each, which is the novelty and contribution of this research.

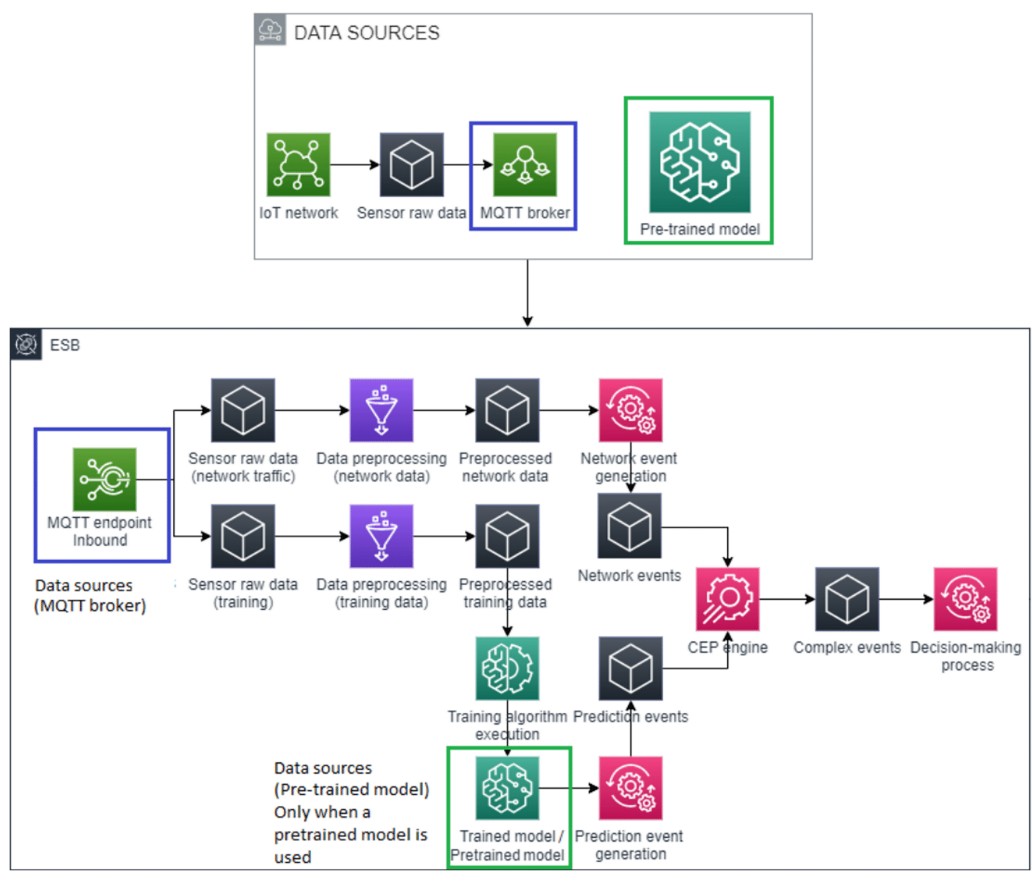

**Figure 1** Generic architecture to detect attacks on IoT devices.

## Architecture proposal

Our proposed architecture for detecting attacks on IoT devices is described below. This architecture, which is an improved version of the architecture we presented in *Roldán et al. (2020)*, is composed of three different parts.

The first module of the architecture, the data sources, consists of the data obtained from the network and the pre-trained model, if available. Otherwise, the model would have to be trained. As shown in Fig. 1, this module may be detached from the rest of the architecture, because it can be replaced by any computer network with an MQTT broker as collector. However, we consider it useful to analyze the whole system to understand its behavior. Note that in this new version of our architecture, an MQTT broker can be used with different topic numbers, with the aim of managing data grouped by type. Additionally, this new version permits the use of pre-trained models as data sources, which allows us to migrate our model from our architecture to other deployments. In addition, pre-trained models provide greater flexibility because they allow training the model outside (or inside) our deployed architecture.

The second module of the architecture, which is in fact the main module, receives raw network data and, optionally, a pre-trained model. This module is responsible for making

decisions on the basis of the network data analyzed in real time. This new version of our architecture is more flexible since different CEP engines can be used according to the user's needs.

At this point, the pipeline of the second module should be explained in detail. Through an MQTT inbound endpoint, the raw network data produced by data sources can reach the ESB. These data are preprocessed to make them consumable for the network event generator. The event generator provides network events which can be received and processed in real time by the CEP engine. Moreover, our architecture needs a trained model to predict the network event values. In particular, this model can be used to predict the type of network packet *via* a predicted value and a threshold, which is computed using the training data. In this case, our model has been built using a linear regression, and is used to predict values and a threshold from a key feature, or features, which is the packet length in our case. These features will vary with each case.

The last module is composed of data sinks which receive the notifications about the decision-making process conducted by the second module. Databases, event systems, emails, logs, or any other system required by end users to receive such notifications are examples of data sinks. Due to its simplicity, an explanatory diagram is not included.

We would like to point out that our architecture allows us to fit the model with raw sensor data; this traffic should be isolated and without any security attacks. There are two ways to obtain prediction patterns: the first is to set a pre-trained model, while the second is to train the model with the isolated network traffic. Regardless of the method which is selected, the architecture uses this model to predict the expected value of each incoming network packet. This prediction is used to create a prediction event which is compared with its corresponding network event. In this way, our architecture is able to obtain patterns which can detect anomalous packets by using the real value, the predicted value and a calculated threshold, since the absolute value of the subtraction of the real value and the predicted value must be smaller than the threshold; otherwise, the packet is anomalous.

Equation (1) describes our predictor in a formal way, where the number 1 means that the network packet belongs to the category used to train the model and obtain the ERROR.

$$f(x) = \begin{cases} 1 & if \quad (abs(real\ Value - predicted\ Value) \leq ERROR) \\ 0 & if \quad (abs(real\ Value - predicted\ Value) > ERROR) \end{cases} \tag{1}$$

It is important to note that we can fit the model with more attacks; for example, if we have traffic from a DoS attack, we can refit our model to detect this attack. The best way to generate patterns is to attack the architecture or obtain traffic from attacks. As mentioned above, we have improved the architecture to accept pre-trained models.

An initial deployment of the architecture could be composed of a few patterns that can be proposed and designed by the domain expert, and the anomaly detector, which uses legitimate traffic. When an anomalous pattern is triggered, anomalous packets can be used

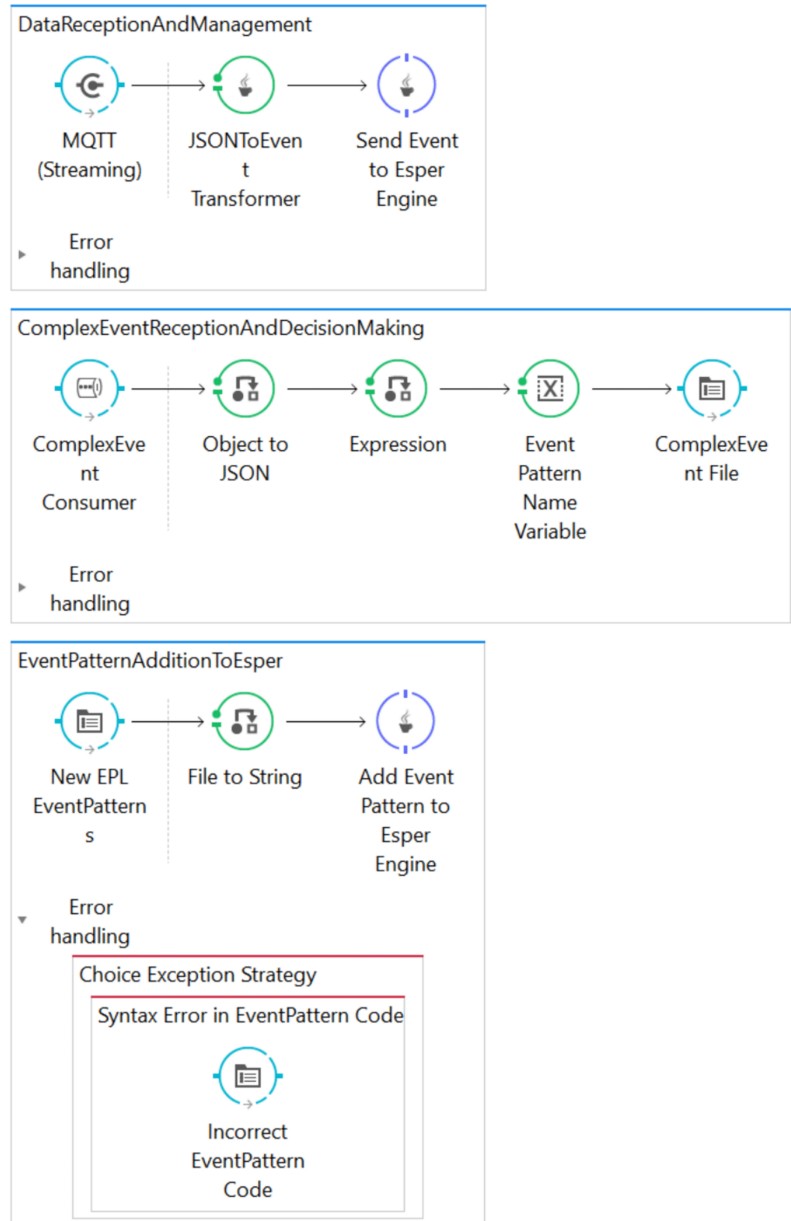

**Figure 2 Screenshot of the implemented Mule-based architecture.**

to generate a new pattern to detect this kind of anomaly again. This means that our architecture can improve and gradually become more accurate over time.

## Architecture implementation with Mule

In this subsection we explain how our architecture for IoT security has been implemented by using the Mule ESB together with the Esper CEP engine.

The Mule-based architecture is composed of three data flows: *DataReception AndManagement*, *ComplexEventReceptionAndDecisionMaking* and *EventPattern AdditionToEsper* (see Fig. 2).
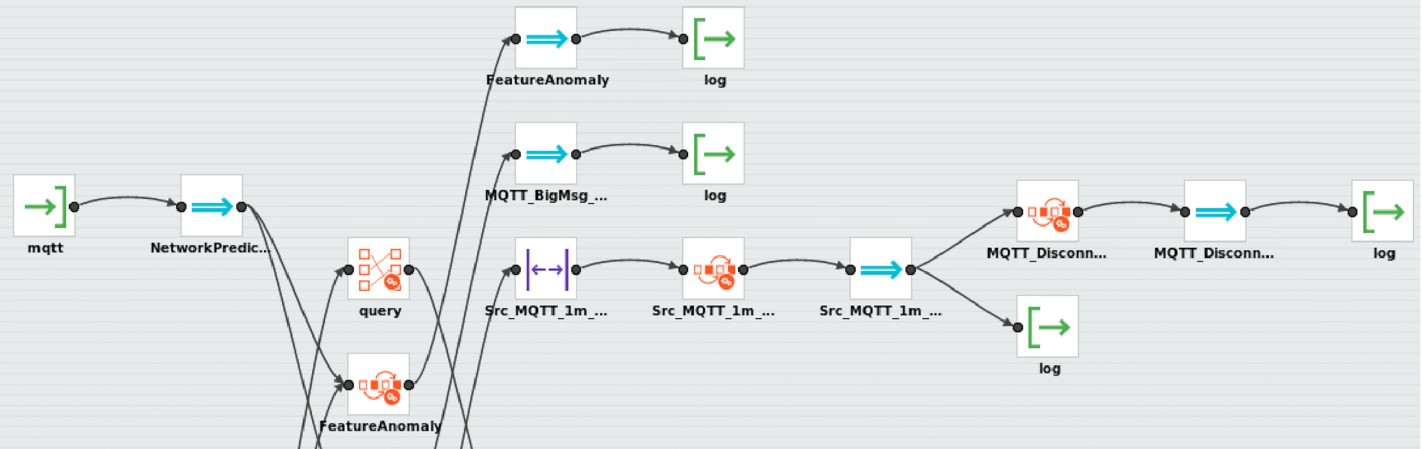

**Figure 3 Screenshot of the implemented WSO2-based architecture.**

The *DataReceptionAndManagement* flow is responsible for receiving data from IoT data sources, transforming them into an event format and then sending them to the Esper CEP engine. Specifically, this flow is implemented with an MQTT inbound endpoint in which a topic is defined to receive the data obtained from data sources. Then, a Java transformer allows the transformation of the received JSON data into Java Map events, which are sent to an Esper CEP engine through a customized message component.

The *ComplexEventReceptionAndDecisionMaking* flow receives the complex events that are automatically generated by the CEP engine upon detection of previously deployed patterns, and transforms these complex events into JSON format. These are then saved in log files, which are a type of data sink for the architecture.

Finally, the *EventPatternAdditionToEsper* flow allows the runtime deployment of new event patterns in the CEP engine. To this end, a file input endpoint frequently checks whether there is a new file with an EPL extension, and if there is the event pattern code contained in this file is transformed into a string, which is then deployed in the Esper CEP engine.

## Architecture implementation with WSO2

Figure 3 depicts our architecture for IoT security that is implemented with the WSO2 ESB. Unlike the implementation of the Mule-based architecture, which was integrated with the external Esper CEP engine, the implementation of the WSO2-based architecture does not require integration with an external CEP engine since WSO2 provides the Siddhi CEP engine by default.

As shown in Fig. 3, the architecture receives the data obtained from data sources (*NetworkPacket* and *NetworkPrediction*) by using an MQTT broker with two topics. Then, these data are matched through the different event patterns (queries) implemented with SiddhiQL and previously deployed in the Siddhi CEP engine. When a complex event is automatically created upon a pattern detection, it is saved in a log file, which is a data sink for the architecture.

# COMPARING ARCHITECTURE PERFORMANCE AND STRESS

This section presents our comparison of the performance and stress tests conducted for the two architectures implemented with Mule and WSO2.

## Proposed approach

Before analyzing each architecture component in depth, a schematic overview of the steps followed to address this comparison are explained below:

- First, a virtualized MQTT network, in which clients publish periodically, is deployed.
- Then, packets are collected from that network to define a *normal* scenario, in which the system is not under attack.
- Afterwards, a malicious client is introduced into the network and this client launches the attacks. Packets that generate attacks are collected to perform the experiments.
- A number of these packets are preprocessed and used to train the linear regression model. The mean square error for each category to be predicted with the regressor is also extracted.
- The values of the packets that were not used for training are predicted and saved. They will be used to perform the experiments.
- Then, event patterns are defined. To perform a complete comparison, we create a pattern per attack that will be able to detect the attack, as a domain expert would do, except for *DoS*, which is detected with a regressor because in practice it is difficult to establish a specific pattern for this type of attack. In addition, we create the *FeatureAnomaly* pattern, which is able to detect anomalies using the linear regressor. This pattern is used to detect unknown attacks, such as *Subfuzzing*, *DoS* or *Discwave*. And then there is the *ProtocolAnomaly* pattern, which detects any unknown protocol that should not be present in the network.
- Both platforms, Mule and WSO2, are deployed with their corresponding patterns.
- The simulator is used to perform the experiments (see next subsection) in such a way that these experiments are reproducible.
- Finally, the metrics of the experiments are extracted for comparison.

## Simulator

To ensure the reproducibility of the experiments, we implemented an MQTT network simulator. We chose an MQTT simulator because, as mentioned above, it is a widely-used protocol in the IoT paradigm. Moreover, MQTT networks, by the nature of the protocol, are usually centralized because the broker acts as a centralizer, so that all MQTT packets pass through it. This makes it very easy to set up a network-based IDS in the broker, because there is no need to redirect traffic to another device. This simulator is capable of sending network packets to an MQTT broker, taking as data source different CSV files which contain real network traffic that was previously generated and stored. This is essential because it allows us to use real traffic and to combine the reproducibility of

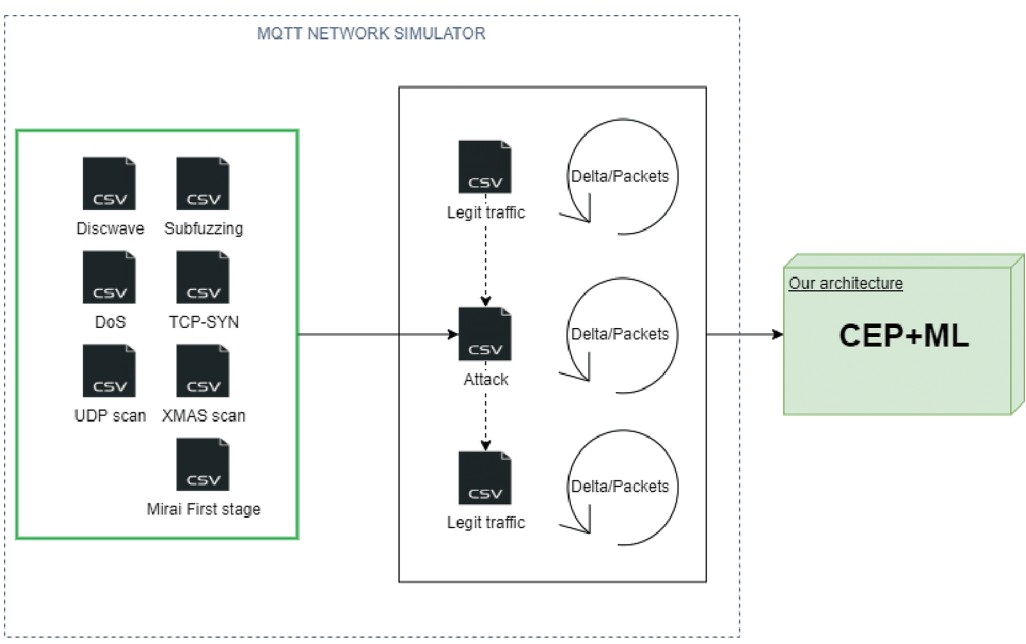

**(a)** MQTT network simulator

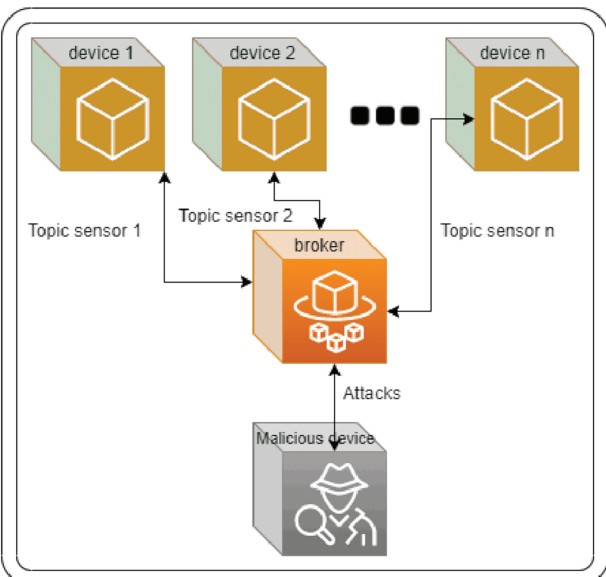

**(b)** MQTT network diagram

**Figure 4 MQTT network (A) and MQTT simulator (B).**

the experiments with data that have been generated in a real MQTT network.

The main advantages of our simulator are that it can reliably send such network packets while taking the delay between packets into account, and it allows us to generate several scenarios to test both the proposed architectures.

Figure 4A outlines this MQTT network simulator. Note that when we wish to generate heavy workloads, we can use the sum of deltas from the packets or the number of packets as the threshold which is used to stop the generation of packets.

The behavior of the simulator is quite simple. First, it reads the CSV files, which first allows us to avoid the delay that is due to reading each row of the CSV while we are sending them.

When the simulator has read both CSV files (legitimate traffic, and the specific attack), it starts to send packets with MQTT, these being sent using JSON format. The number of packets is defined as described above.

Figure 4B shows an MQTT network diagram where there are a certain number of legitimate devices, 4 in our case, and 1 malicious device which attacks the network in different ways. This network is similar to the network used to obtain the network traffic.

## Event patterns

In our previous work (*Roldán et al., 2020*), we defined and implemented twelve event patterns in Esper EPL for detecting the following security attacks:

- *TCP/SYN port scan*: the malicious device sends a round of 10 or more TCP packets with the SYN flag to three or more different ports of the broker in 1 s. If the port is open, the broker sends a SYN/ACK packet, otherwise it sends an RST packet. The *TCP_SYN* pattern implements this attack by making use of an intermediate pattern called *SrcDst_TCP_1s_Batch*.

- *UDP port scan*: the malicious device sends a round of 10 or more empty UDP packets to three or more different ports of the broker in 5 s. If the broker sends any response, then the port is open, but if the broker does not send a response, the port could be open. If the broker sends ICMP unreachable, the port should be closed. And if it sends a different error (not unreachable), the port should be filtered. The *UDP_Port_Scan* pattern implements this attack by making use of an intermediate pattern called *SrcDst_UDP_5s_Batch*.

- *Xmas port scan*: the malicious device sends a round of 10 or more TCP packets with PSH, FIN and URG flags to 2 or more different ports of the broker in 1 s. If the broker does not respond, the port should be open or filtered. If the broker sends an RST packet, it should be closed. If the broker sends an ICMP unreachable error, it should be filtered. The *Xmas_Scan* pattern implements this attack by making use of an intermediate pattern called *SrcDst_Xmas_1s_Batch*.

- *TELNET Mirai*: the malicious device simulates the first stage of Mirai, sending connect packets with different tuples (username/password). The pattern can be detected if the attacker sends more than 5 TELNET packets in 1 min. The *TELNET_Mirai* pattern implements this attack by making use of an intermediate pattern called *Src_TELNET_1m_Batch*.

- *MQTT disconnect wave*: the malicious device sends many MQTT packets with the *connect command*. As sending more than 1 *connect command* is strange, the pattern can be detected if the broker receives more than 5 MQTT *connect commands* in 1 min from a single IP address. The *MQTT_Disconnect_Wave* pattern implements this attack by making use of an intermediate pattern called *Src_MQTT_1m_Batch*.

- *MQTT subscription fuzzing*: the malicious device tries to subscribe to all topics, so the pattern can be detected if an MQTT client subscribes to more than 20 topics in 5 min. The *MQTT_Subscription_Fuzzing* pattern implements this attack by making use of an intermediate pattern called *Src_MQTT_5m_Batch*.

In the present work, we have used these twelve event patterns implemented in Esper EPL to test the Mule-based architecture. Moreover, we have implemented analogous patterns but in SiddhiQL to test the WSO2-based architecture.

Additionally, in this work we have split the *Anomaly* pattern, proposed in *Roldán et al. (2020)*, into 2 new patterns: *ProtocolAnomaly* and *FeatureAnomaly*. The first pattern allows us to detect protocols which are not expected because this may suggest that the system is under attack. The second pattern allows us to detect anomalous packets in expected protocols. Thus this pattern split allows us to classify anomalies more accurately.

Listing 1 shows the *FeatureAnomaly* pattern implemented in Esper EPL, while Listing 2 contains the implementation of the same pattern but using the SiddhiQL language. This pattern implements Eq. (1) and allows us to detect unmodeled attacks, such as the *DoS with big messages*. Moreover, it will detect other attacks, such as disconnect wave or subscription fuzzing, even if we do not define specific patterns to detect them. The *ProtocolAnomaly* pattern implemented in Esper EPL is shown in Listing 3, while Listing 4 contains the same pattern using the SiddhiQL language.

We have implemented two types of event patterns to detect such attacks. The first type uses a time batch window (*SrcDst_TCP_1s_Batch*, *SrcDst_UDP_5s_Batch*, *SrcDst_Xmas_1s_Batch*, *Src_TELNET_1m_Batch*, *Src_MQTT_1m_Batch* and *Src_MQTT_5m_Batch*) to trigger a complex event when a condition is met. The second type of pattern allows the comparison of messages coming from two broker topics, one that manages prediction and threshold data while the other topic manages real packet information. In this case, the pattern is activated when the difference between the prediction and the real values is higher than a certain threshold; this is useful because we can compare the performance for different attacks but also with different types of patterns.

## Machine learning model

Selecting a machine learning model is a very important step in effectively deploying the architecture. Although this paper does not focus on ML processes, it is important to give a brief explanation of the model we have used.

The first step in defining our ML model was to select the most important features. For this purpose we applied the criteria proposed by KDD99, which are adaptable to our MQTT dataset. In addition, we also added features obtained from MQTT.

Once the features have been selected, they are normalized and binarized when necessary. Then we used Extremely Randomized Trees with our dataset to arrange the features by importance. After that, we selected the most important features (*Geurts, Ernst & Wehenkel, 2006*).

Table 1 show the importance of the binarized features. One or several features are chosen to be the key feature/s, and these are predicted with the rest of the features

**Listing 1** *FeatureAnomaly* pattern implemented in Esper EPL.

```
@Name("FeatureAnomaly")
@Tag(name="domainName", value = "IoTSecurityAttacks")
insert into FeatureAnomaly select a2 . id as id,
current time stamp( ) as time stamp, a1 . destIp as destIp
from pattern[((every a1 = NetworkPacket((a1 . protocol = 'MQTT' or
a1 . protocol = 'TCP')))
-> a2 = NetworkPrediction((a2 . i d = a1 . id and
(a2 . packetLengthPredict < (a1 . packetLength
- a2 . packetLengthPredictSquaredError) or a2 . packetLengthPredict >
(a1 . packetLength
+ a2 . packetLengthPredictSquaredError))))))]
```

**Listing 2** *FeatureAnomaly* pattern implemented in SiddhiQL.

```
@info(name= "FeatureAnomaly")
from ((every a1 = NetworkPacket[(a1 . protocol == 'MQTT'
or a1 . protocol == 'TCP')]) -> a2
= NetworkPrediction[(a2 . i d == a1 . id and
(a2 . packetLengthPredict < ( a1 . packetLength
- a2 . packetLengthPredictSquaredError) or
a2 . packetLengthPredict > (a1 . packetLength
+ a2 . packetLengthPredictSquaredError)))])
select a2 . id a s id,
time : timestampInMilliseconds( ) as time stamp,
a1 . destIp as destIp
insert into FeatureAnomaly;
```

obtained. Furthermore, this prediction will be compared with the real value for each event, with the error threshold being defined using the mean square error obtained when we train the model.

By using pre-processed features, we can select the model. In this case, our data features fit a linear distribution very well. Therefore, we chose a linear regression to predict these key features. This model can change depending on the whole IoT network.

## Tests

By implementing a network simulator, we were able to measure the performance of our proposed architecture implemented with WSO2 and Mule, and compare them. We designed 14 experiments with seven different attacks against MQTT, and each test was composed of legitimate traffic and one specific attack. Specifically, we carried out seven

**Listing 3** *ProtocolAnomaly* pattern implemented in Esper EPL.

```
@Name("ProtocolAnomaly")
@Tag(name="domainName", value= "IoTSecurityAttacks")
insert into ProtocolAnomaly
select a1.id as id,
current_timestamp() as timestamp,
a1.destIp as destIp
from pattern[(every a1 = NetworkPacket((a1.protocol != 'TCP' and
a1.protocol != 'UDP'
and a1.protocol != 'MQTT' and
a1.protocol != 'ARP' and a1.protocol != 'DHCP'
and a1.protocol != 'MDNS' and
a1 . protocol != 'NTP' and a1.protocol != 'ICMP'
and a1.protocol != 'ICMPv6' and
a1.protocol != 'DNS' and a1.protocol != 'IGMPv3')))]
```

**Listing 4** *ProtocolAnomaly* pattern implemented in SiddhiQL.

```
@info(name="ProtocolAnomaly")
from (every a1 = NetworkPacket[(a1.protocol != 'TCP' and
a1.protocol != 'UDP'
and a1.protocol != 'MQTT' and
a1.protocol != 'ARP' and a1.protocol != 'DHCP'
and a1.protocol != 'MDNS' and
a1.protocol != 'NTP' and a1.protocol != 'ICMP'
and a1.protocol != 'ICMPv6' and
a1.protocol != 'DNS' and a1.protocol != 'IGMPv3')])
select a1.id as id,
time:timestampInMilliseconds() as timestamp,
a1.destIp as destIp
insert into ProtocolAnomaly;
```

experiments (one per attack) which used the delay of each packet in order to simulate a network realistically, and seven experiments without a delay, which allowed us to measure the performance with heavy workloads. Thus, the proposed tests were as follows:

- TCP-SYN scan (with delay/without delay)
- UDP port scan (with delay/without delay)
- XMAS port scan (with delay/without delay)

- Mirai first stage (with delay/without delay)
- MQTT disconnect wave (with delay/without delay)
- MQTT subscription fuzzing (with delay/without delay)

## RESULTS

This section presents and discusses the experiments and the results obtained when comparing the performance of our architecture implemented with WSO2 and Mule, as well as the limitations of each implementation.

These experiments were carried out under similar conditions for the WSO2-based architecture, composed of the WSO2 ESB and the WSO2 CEP engine, and the Mule-based architecture that integrates Mule ESB with the Esper CEP engine. We would like to point out that WSO2 provides some extra performance features such as multiworkers and PMML models(*WSO2, 2020*, *2021a*), which could enhance the architecture's performance. However, we did not integrate these features in our proposed architecture in order to create similar conditions for both systems.

The results obtained for the two types of tests conducted in this work (performance and stress tests) are discussed below. The implementation code can be accessed in the *Roldán-Gómez et al. (2021)* repository.

### Performance tests

The results for the performance tests are presented in the following subsections.

#### *Estimated computational complexity*

Although it is difficult to give an exact figure for computational complexity because of the internal operations performed by the CEP engines, we estimate the computational complexity on the basis of the steps that we can calculate. Note that this estimation assumes that the model and preprocessing steps are as mentioned. Obviously, this will change if another model or steps are used during the preprocessing step.

To define the computational complexity, we consider the following variables: $n$, which defines the number of packets, which $F$ is the number of variables where each step is applied (this value will be constant for each step); and $v$, which defines the different values of each category and is used only for the binarization of categorical attributes. First, we calculate the computational complexity of each step, then the total for the training stage, and then the total at runtime.

The estimated computational complexities are as follows: min-max scaler $O(2nF_1)$, fill empty values $O(nF_2)$, binarization of categorical attributes $O(2nF_3v)$, and training linear regression model $O(nF_4^2 + F_4^3)$. All these steps only have to be carried out once. In addition: predict a value with the regressor and create $n$ events $O(F_5n)$. In summary, the estimated computational complexity in training is as follows:

$$O(2nF_1 + nF_2 + 2nF_3v + nF_4^2 + F_4^3).$$

| Table 1 Feature importance. | |
|---|---|
| **Feature name** | **Feature importance** |
| Destination port (1883) | 0.259 |
| Calculated window size | 0.240 |
| Protocol (TCP) | 0.122 |
| Protocol (MQTT) | 0.100 |
| IP source (192.168.1.11) | 0.092 |
| Information (Publish message) | 0.032 |
| Source port (59662) | 0.030 |
| IP source (192.168.1.7) | 0.029 |
| Source port (62463) | 0.027 |
| Source port (52588) | 0.016 |
| Packet length | 0.005 |

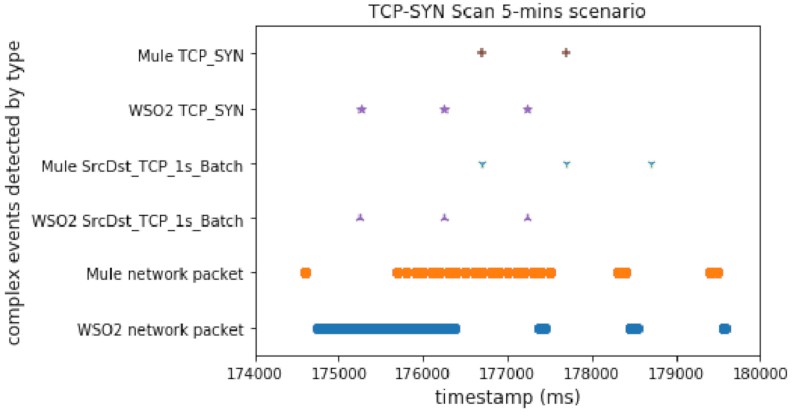

**Figure 5 TCP_SYN attack comparison.**

And the estimated computational complexity at runtime is:

$$O(F_5 n).$$

Since we do not know the exact inner workings of CEP engines, it is difficult to calculate the remaining steps. That is why performance experiments, such as those carried out in this paper, are so important.

### TCP SYN scan

The first experiment performed was composed of legitimate traffic (a simple MQTT network) and a TCP SYN scan. We used our architecture as an IDS to detect attacks or scans.

Figure 5 shows the results obtained for the TCP-SYN scan test executed for 5 min on both the WSO2-based architecture and the Mule-based one. The X-axis represents the execution time in milliseconds, while the Y-axis shows the different complex event types detected in real time during the execution of the test.

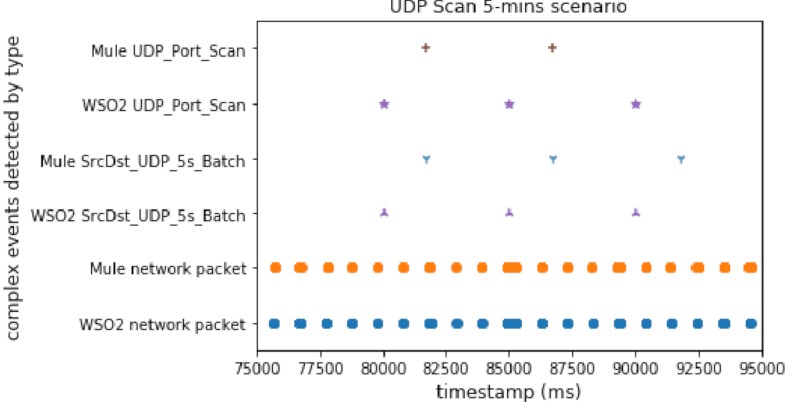

**Figure 6** UDP scan attack comparison. 

As we can see, the WSO2 implementation triggers the TCP_SYN complex event first. Therefore, we can conclude that the WSO2 achieves an earlier detection than the Mule-based one. In this case, TCP_SYN starts sooner in the WSO2 scenario, but this delay is shorter than the detection time difference.

### UDP port scan

The UDP scan is slower than the TCP one, and it is useful to study the performance in a different way. This experiment allowed us to compare the performance when the attack has a low packet sending ratio. As in the case of the TCP SYN Scan experiment, there was normal traffic and a UDP port scan.

Figure 6 shows the results obtained for this UDP port scan experiment. In conclusion, we can say that WSO2 was faster than Mule again. Mule generated a null window, not being able to detect the third *UDP_Port_Scan* complex event. It is important to note that the difference is smaller than in Fig. 5; this may be because the attacks, in both cases, started at the same time.

### Xmas port scan

This scan is not very common and shows how our architecture is able to detect more unusual attacks. From the point of view of the experiment, it should be like the TCP_SYN scan, as both have similar packet sending ratios and event generation characteristics.

Figure 7 shows that WSO2 was faster than Mule, even though the Xmas port scan attack started sooner in the Mule scenario. This experiment is useful because it allowed us to confirm the superiority of WSO2 when there is not a comparison between different events.

### Mirai first stage

This scenario simulates the first stage of Mirai. This attack tries to connect with Telnet using a username/password list. The main aim of this experiment was to check the behavior of our system under common IoT attacks. Figure 8 shows a comparison of the results for the Mirai scenario, executing it on the WSO2-based architecture and the ESB-based one. Again, WSO2 detected the first complex event faster than Mule.

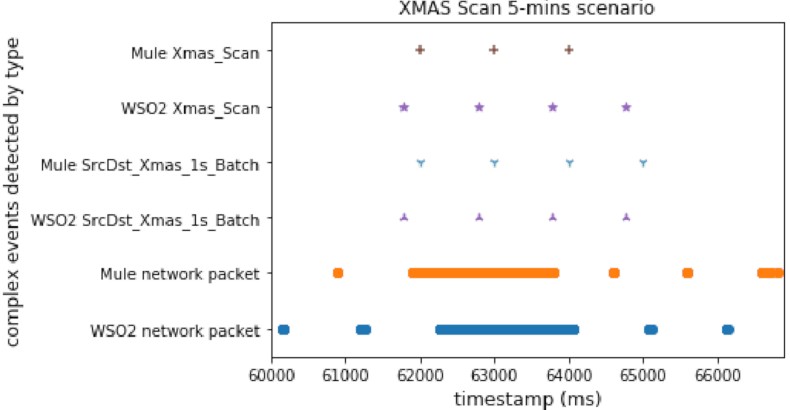

**Figure 7  XMAS scan attack comparison.**     

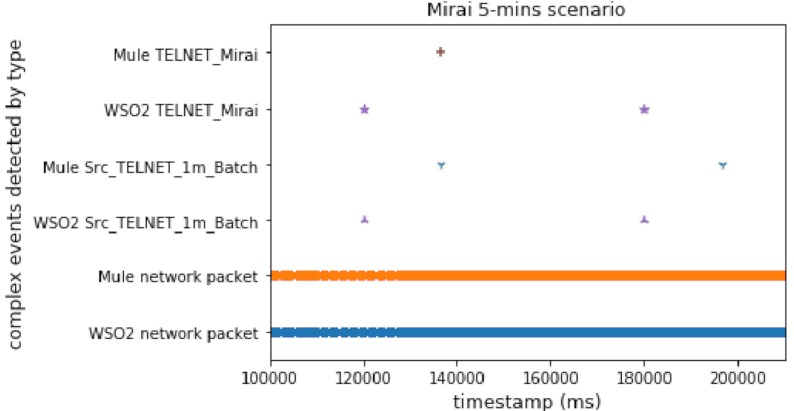

**Figure 8  Telnet-Mirai attack comparison.**     

### DoS big message

This scenario simulates a common DoS attack in which the attacker sends big messages quickly to the broker.

This experiment is different to the other ones because time windows are not used. Instead of time windows, each packet is matched with its prediction. As we mentioned above, there are two different ways to detect attacks using our predictor. In this case, the system trains the model with legitimate and isolated traffic, allowing us to detect anomalous packets. Note that each packet which does not match with its prediction, and whose difference exceeds the threshold, can be classified as anomalous. Additionally, we could have fitted a model to detect each specific attack.

Figure 9 illustrates that, in this case, Mule was faster than WSO2, since WSO2 needed more time to detect all the malicious packets. Therefore, we can conclude that Mule offers better performance when we need to compare different events (network events and prediction events in this case).

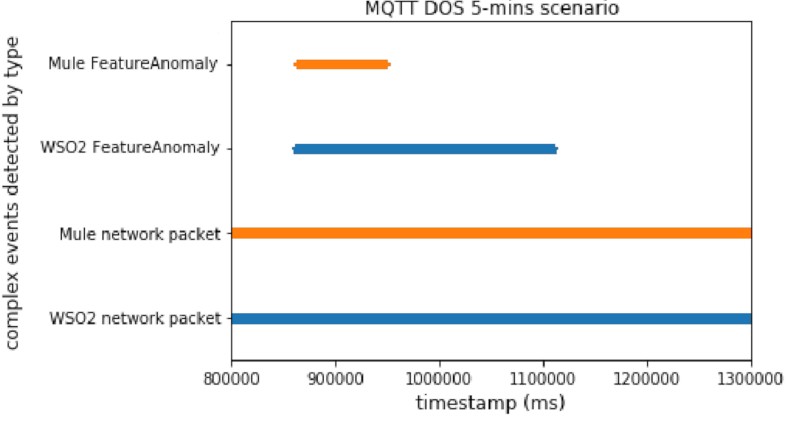

**Figure 9** **DoS big message attack comparison.**

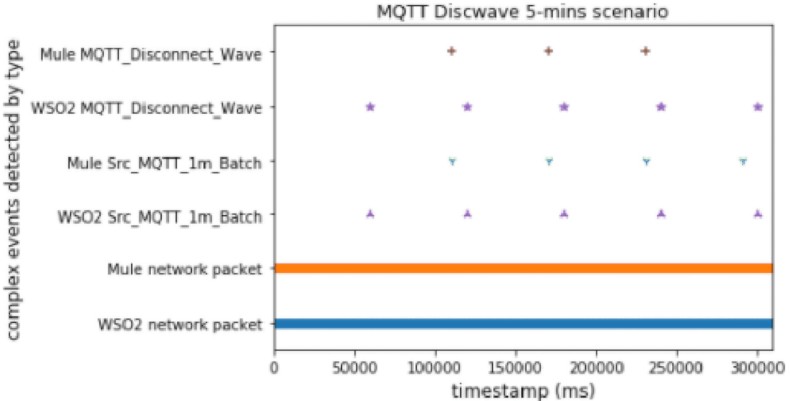

**Figure 10** **Discwave attack comparison (using time windows).**

### MQTT disconnect wave

This scenario provides useful knowledge about both platforms. Here there are time windows and an anomalous packet detector, which works by matching each packet with its prediction, as we did in the DoS experiment. The advantage of this experiment is that it allowed us to check the behavior of the whole proposal deployed with the predictor working. Note that in a real scenario we would not use both methods (time windows and prediction), but it was useful and appropriate to test the performance.

As we can conclude from Figs. 10 and 11, Mule again worked better with predictions (by using two topics) than WSO2. On the other hand, WSO2 again detected the first complex event earlier than Mule when using time windows.

### Subscription fuzzing

In this scenario, we used both methods again (time windows and predictions), but this attack is slower than the discwave one, which meant that the delay between packets was longer than in the discwave attack. This experiment shows the behavior of our

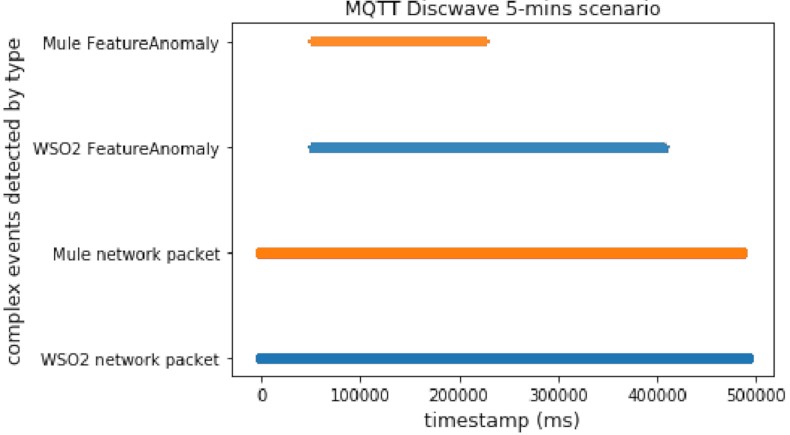

**Figure 11 Discwave attack comparison (using FeatureAnomaly pattern).**

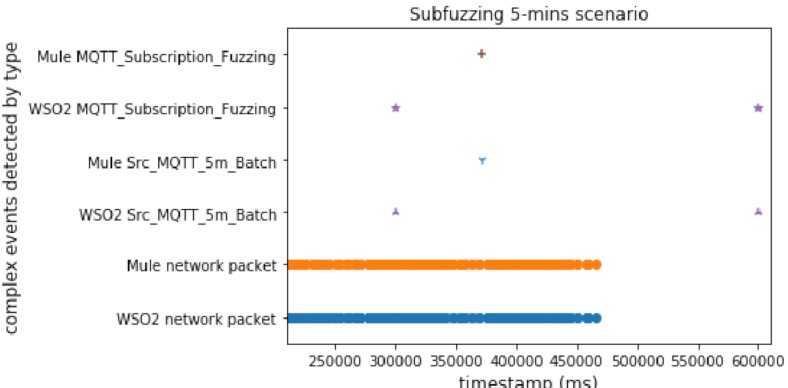

**Figure 12 Subfuzzing attack comparison (using time windows).**

proposal when the system receives an attack with a lower packet sending ratio than DoS or discwave.

Figures 12 and 13 show that there are two interesting facts that we can extract from this experiment. The first is that WSO2 detected the second complex event very late when using time windows. As it uses a 5-min window, the second time window was closed after the attack finished. But the important thing is that, in this case, WSO2 and Mule presented a similar performance with predictions. This is due to the long delay between packets in this experiment. WSO2 again registered the first detection sooner. It seems that Mule was processing a heavy workload when we compared two different events, but WSO2 provided a better brute performance when the system compared features/properties in the same event.

## Stress test

Additionally, we carried out 7 more experiments in which the network packets had no delay. Although this is not a realistic case, it is very useful because it allows us to study the

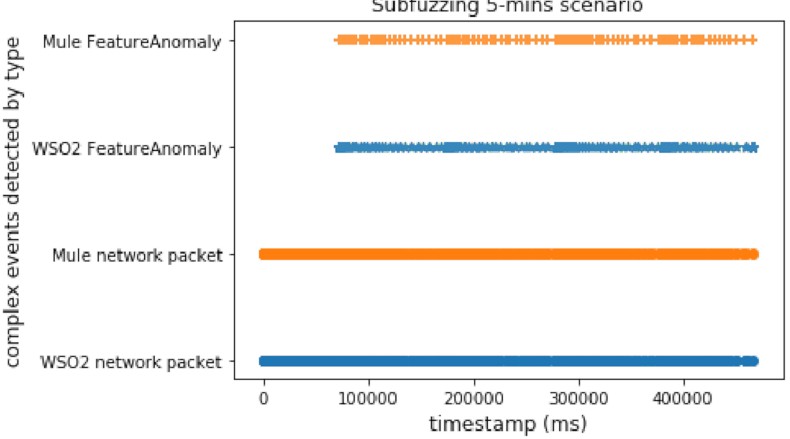

**Figure 13 Subfuzzing attack comparison (using FeatureAnomaly pattern).**

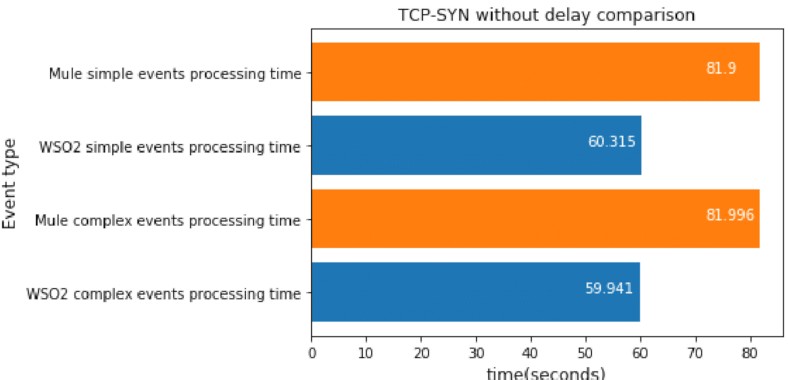

**Figure 14 TCP-SYN without delay comparison.**

difference in performance between the two architecture implementations in greater depth. The figures in this subsection compare the last simple event detected with the first one, as well as the last complex event detected with the first one, measuring the time differences between these events.

### TCP-SYN without delay

For each attack mentioned above, we implemented a stress scenario.

In this case, we executed the TCP-SYN scan 100 times, which took about 1 min.

Figure 14 shows the difference between the last and the first simple events detected, as well as that for the last and first complex events detected. Our goal was to discover the processing speed difference between the platforms.

As we can see, WSO2 was faster at processing simple events and complex events than Mule when there was a single broker topic, so this experiment confirms the results obtained in the previous section. It seems that, regardless of the packet delay, WSO2 is faster at processing simple events and complex events when there are no relationships between them.

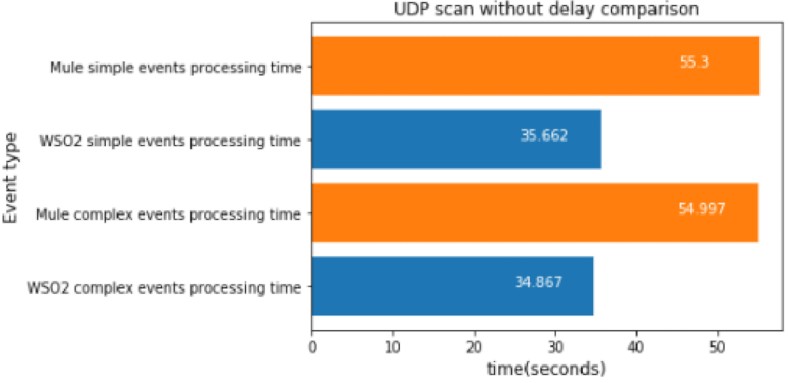

**Figure 15 UDP scan without delay comparison.**

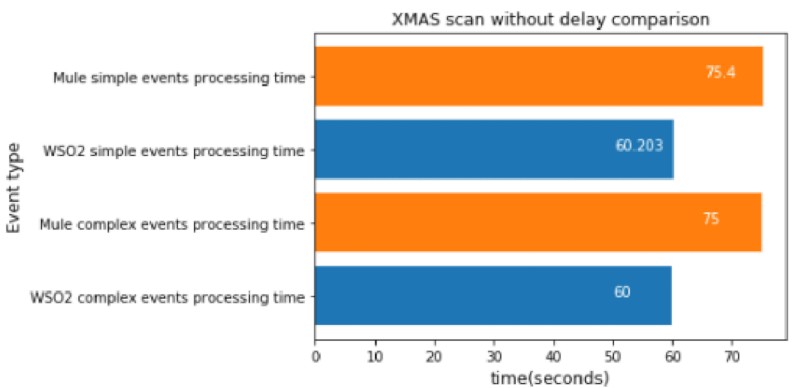

**Figure 16 XMAS Scan without delay comparison.**

### UDP scan without delay

In this case, the UDP scan was launched 100 times, which took about 37 s.

Figure 15 depicts a similar result to the TCP-SYN experiment: WSO2 was faster again. It is interesting that the differences in the experiments without a delay between WSO2 and Mule are far bigger than in those with a delay; this is because these experiments generate many more events than the experiments with a delay.

### XMAS scan without delay

The XMAS scan was executed 100 times again, which took about 60 s.

As we can see in Fig. 16, the results are consistent with those we have observed above. In this experiment, WSO2 was faster again.

### Mirai first stage without delay

The first stage of Mirai was executed 100 times, which took about 48 s.

Figure 17 shows that WSO2 was faster again at processing simple events.

### DoS big message without delay

The DoS scenario does not use time windows, instead it compares each packet with its prediction. We executed the DoS experiment without delay once, which took about 20 s.

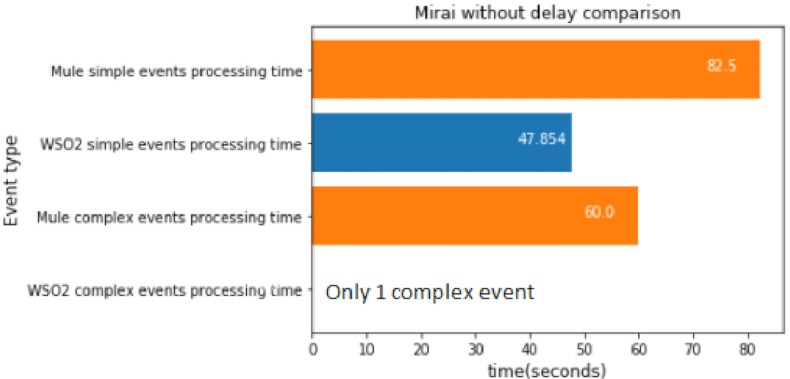

**Figure 17 Mirai first stage without delay comparison.**

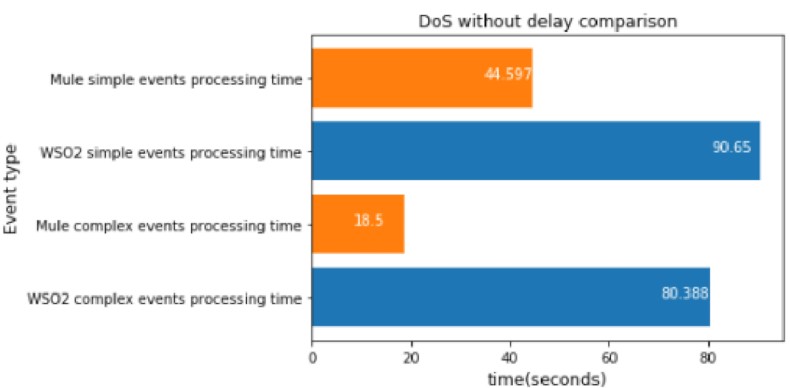

**Figure 18 DoS without delay comparison.**

The results are very interesting, as illustrated in Fig. 18. They show that Mule was faster than WSO2 when there was an operation between 2 broker topics. The performance difference between implementations was even bigger than in the experiments with one type of simple event.

### MQTT disconnect wave without delay

We executed the discwave attack for about 27 s, and used the *FeatureAnomaly* prediction pattern to detect it.

Fig. 19 shows that Mule was faster again when we compared 2 different events. This experiment had the highest workload, and therefore the difference between WSO2 and Mule was even bigger than before.

### MQTT subscription fuzzing without delay

This experiment consisted in running the subfuzzing attack for approximately 47 s.

As we can see in Fig. 20, Mule was much faster again, so we can conclude that WSO2 is only faster than Mule when there are no comparison operations between different events.

In short, each CEP engine has different advantages. The Esper CEP engine integrated with the Mule ESB is better when there are comparisons between different events, so

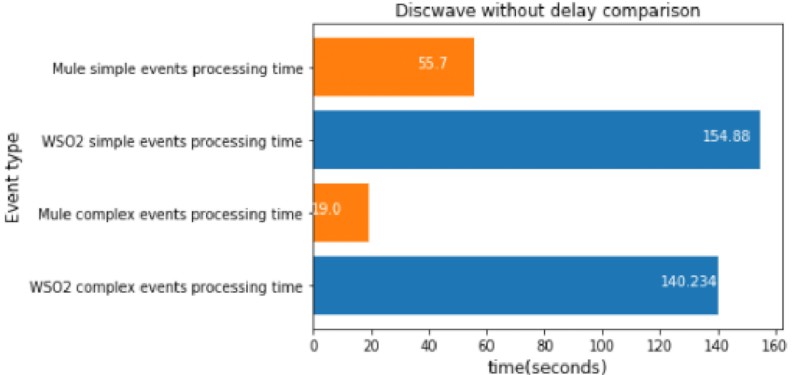

**Figure 19 Discwave without delay comparison.**

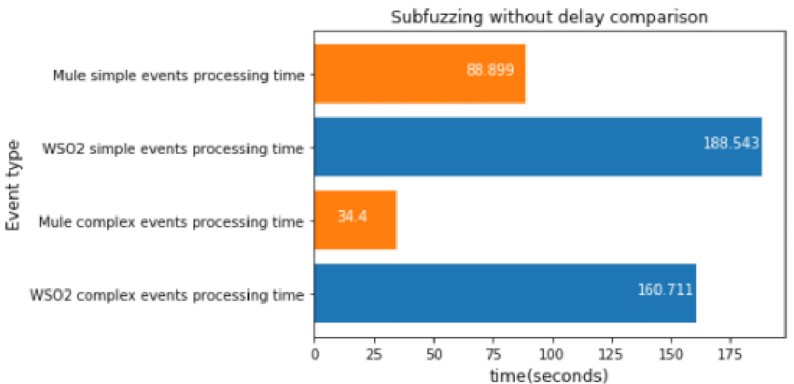

**Figure 20 Subscription fuzzing without delay comparison.**

Esper/Mule performs better on patterns where different events are compared. As an example, we can see this behavior in the anomalous packet pattern. However, when there are no comparisons between different events, WSO2 is faster than Esper/Mule. We can conclude that WSO2 provides a better raw performance, in other words, WSO2 is able to process network packets faster than Esper/Mule but its performance is worse when there are comparisons between events.

## DISCUSSION

With the obtained results, we can discuss and answer the four research questions posed in *Introduction* section:

### Answers to the Research Questions

- **RQ1**: Can a real-time data stream processing architecture be implemented with the WSO2 ESB together with the Siddhi CEP engine and be integrated with ML techniques?

  – We can definitely affirm that it is possible to implement a streaming data processing architecture using WSO2 ESB together with the Siddhi CEP engine and integrate them with ML techniques. In fact, we have implemented an architecture equivalent to

the one presented in *Roldán et al. (2020)*, but using the said WSO2 technologies. We have also tested its functionality in a realistic environment consisting of security attacks in the field of the IoT.

- **RQ2**: Can a streaming data processing architecture based on the integration of ML techniques with the WSO2 CEP engine and ESB achieve or improve upon the performance of the previously proposed architecture (*Roldán et al., 2020*)?

  – We can undoubtedly say that WSO2 CEP and ESB can achieve a performance similar to that achieved by integrating Esper CEP and Mule ESB in an equivalent streaming data processing architecture for detecting security attacks in the IoT. We have carried out a series of tests with a number of typical attacks on communication protocols in the IoT environment, and we have seen that both architectures achieve an appropriate and similar performance, although we did detect that each of them can achieve a better performance with certain types of patterns, which allows us to answer our next research question.

- **RQ3**: What kind of event patterns are processed faster with WSO2/Siddhi and which ones with Mule/Esper, and which of the two architectures is more suitable for supporting high-stress situations?

  – On the one hand, we have observed that the WSO2-based architecture is faster at processing simple events when there are no pattern comparisons between different event types. This is because WSO2 has a higher performance when processing simple events. On the other hand, the Mule-based architecture has shown to be faster when comparing different types of events. The behavior of the architectures under stress will depend on the type of pattern conditions. If we are able to avoid patterns with comparisons between events of different types, WSO2 will be faster in a high-stress situation, since its ESB has a higher performance when processing simple events. Otherwise, Mule will be faster.

- **RQ4**: Which of these architecture implementations is the best to be deployed in an IoT security attack detection environment?

  – Both implementations are effective, but in this context we advocate the choice of WSO2 because it allows us to integrate the different types of events in a general unified event. This dramatically increases the performance of WSO2. Both ESBs can be deployed in an IoT environment, but WSO2 is faster when using this general event (as we can see from the stress experiments). Despite this, Mule can be deployed successfully too, but its performance is worse than that of WSO2.

## CONCLUSIONS AND FUTURE WORK

This paper has presented and compared two implementations of an intelligent SOA 2.0-based architecture integrated with CEP technology and ML techniques that are designed to

detect security attacks against IoT systems. Each of the implementations incorporates a CEP engine and an ESB from prestigious vendors: Esper CEP and Mule ESB on the one hand, and WSO2 ESB and Siddhi CEP on the other.

The validation process, through which the behavior of both architectures was evaluated under the same conditions in a realistic scenario of security attacks on IoT protocols, allowed us to draw the following relevant conclusions:

- Both implementations of the architecture allow us to detect well-known attacks in the field of IoT protocols, with the corresponding event patterns of these attacks.
- Thanks to the use of ML techniques, the architecture can detect novel attacks that have not previously been defined through specific event patterns.
- Our architecture is able to work as a pure rule-based IDS with patterns defined by an expert, as well as allowing the addition of patterns for detecting non-modeled attacks in order to act as an anomaly detection architecture.
- Both architecture implementations present a suitable degree of efficiency for the field of security attacks in the IoT, but each one has its own advantages and drawbacks.
- The Mule-based architecture is faster when the architecture makes use of 2 message broker topics to compare the values of their features.
- The WSO2-based architecture is faster when there is a single topic and the system has a heavy workload.
- To mitigate the performance degradation, suffered by the system under heavy workloads, the operations between the topics can be modified by joining the prediction and network packet data in a general topic, thus mitigating this problem when comparing 2 topics in the WSO2-based architecture.
- In the Mule-based architecture it is more difficult to overcome this problem because our experiments have shown that the performance of Mule does not improve when there is a single type of topic.

Although our work achieved the proposed objectives, there are certain limitations in specific contexts. One is that although, the architecture makes it possible to define a threshold automatically, it is still necessary to perform a feature selection process. Another is that despite the fact that the architecture is capable of defining a threshold for one or more features, it is not able to fully generate the pattern.

As future work, we plan to test our architecture in a different network to validate our proposal with other protocols and conditions. We would like to point out that the performance of our proposal is subject to a correct ML process (data extraction, data preprocessing, algorithm selection, etc.). It would also be of interest to implement the architecture with additional ESBs and CEP engines to extend the comparison with the products of other vendors. Another interesting line of future work would be to automate the process of feature selection, as proposed in *Wajahat et al. (2020)*, thus providing useful information for the selection of the machine learning model with different

underlying structures in network traffic. These modifications should solve the current limitations of the architecture mentioned above.

### Funding

This work was supported by the Spanish Ministry of Science, Innovation and Universities and the European Union FEDER Funds [grant numbers FPU 17/02007, RTI2018-093608-B-C33, RTI2018-098156-B-C52 and RED2018-102654-T]. This work was also supported by the JCCM [grant number SB-PLY/17/180501/ 000353] and the Research Plan from the University of Cadiz and Grupo Energetico de Puerto Real S.A. under project GANGES [grant number IRTP03' UCA]. The funders had no role in study design, data collection and analysis, decision to publish, or preparation of the manuscript.

### Grant Disclosures

The following grant information was disclosed by the authors:
Spanish Ministry of Science, Innovation and Universities and the European Union FEDER Funds: FPU 17/02007, RTI2018-093608-B-C33, RTI2018-098156-B-C52 and RED2018-102654-T.
JCCM: SB-PLY/17/180501/ 000353.
Research Plan from the University of Cadiz and Grupo Energetico de Puerto Real S.A. under project GANGES: IRTP03_UCA.

### Competing Interests

The authors declare that they have no competing interests.

### Author Contributions

- José Roldán-Gómez conceived and designed the experiments, performed the experiments, analyzed the data, performed the computation work, prepared figures and/or tables, authored or reviewed drafts of the paper, and approved the final draft.
- Juan Boubeta-Puig conceived and designed the experiments, analyzed the data, performed the computation work, prepared figures and/or tables, authored or reviewed drafts of the paper, and approved the final draft.
- Gabriela Pachacama-Castillo performed the computation work, authored or reviewed drafts of the paper, and approved the final draft.
- Guadalupe Ortiz conceived and designed the experiments, analyzed the data, prepared figures and/or tables, authored or reviewed drafts of the paper, and approved the final draft.
- Jose Luis Martínez conceived and designed the experiments, analyzed the data, prepared figures and/or tables, authored or reviewed drafts of the paper, and approved the final draft.

## Data Availability

The data is available at Mendeley: Roldán-Gómez, José; Boubeta-Puig, Juan; Pachacama-Castillo, Gabriela; Ortiz, Guadalupe; Martínez, José Luis (2021), "Dataset for Detecting Security Attacks in Cyber-Physical Systems: A Comparison of Mule and WSO2 Intelligent IoT Architectures", Mendeley Data, V1, doi: 10.17632/fvb9pp5xsh.1.

The code is available as a Supplemental File and the code and patterns are available at GitHub: https://github.com/josE4roldan/Detecting-security-attacks-in-cyber-physical-systems-a-comparison-of-mule-and-WSO2-intelligent-IoT.

## Supplemental Information

Supplemental information for this article can be found online at http://dx.doi.org/10.7717/peerj-cs.787#supplemental-information.

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
