# Peer review of "Detecting security attacks in cyber-physical systems: a comparison of Mule and WSO2 intelligent IoT architectures"

_PeerJ Computer Science, doi:10.7717/peerj-cs.787_

## Round 0.1 · original submission · Major Revisions

The Research Paper needs Major Revisions and will require a re-review before a final decision can be made.

·

Basic reporting

Paper theme is good and it will attract attention in community.

Experimental design

Experimental process is explained in proper way.

Validity of the findings

Validation is done with help of comparative analysis.

Additional comments

1. There is no link of code in the paper. Add github or any other relevant link so that proposed work may be tested.
2. Highlight all assumptions and limitations of your work.
3. Mention all figures properly in text.
4. Mention time complexity of proposed pipeline.

·

Basic reporting

This review is for the paper to be published in PeerJ Computer Science #61593. The title of the manuscript is: “Detecting security attacks in cyber-physical systems: a comparison of mule and WSO2 intelligent IoT architectures”, written by José Roldán-Gómez, Juan Boubeta-Puig, Gabriela Pachacama-Castillo, Guadalupe Ortiz, José Luis Martínez.

The authors wrote the article in good professional English. As the reader, I can easily follow the storyline with ease and clarity. I believe that the article conforms to scientific standards and follows courtesy and expressions.

In the introduction in lines 36,37, I am afraid I have to disagree with the authors about the lack of discussion in the IoT securities space. In my opinion, IoT securities have been discussed and published extensively. The authors should provide more evidence to convince the reader about this statement and how this research contributes novelty to the IoT securities domain.
The literature review is unusually described towards the paper's final section, which is after the results and discussions(lines 661). Usually, the article's structure would have a prior literature review conducted to determine the proper methodology for this purpose. However, the literature review being presented is to compare the performance analysis of this research. I understand what the authors intend to do. By all means, it is not a wrong thing to do, but I would like to suggest the authors should also provide a literature review in order to support the chosen research method. For example, when the authors compare the architecture implementation with Mule and Esper versus WSO2 and Siddhi, the authors should provide broader literature to convince the reader that the architecture being discussed is better than other architecture. In line 343, the author should justify the use of the MQTT network simulator. How is this simulator a better choice compared to other network simulators?

In my opinion, the article already includes an adequate introduction. The background section demonstrates the broader space of knowledge that fits into this research. The authors should provide more initial relevant literature to support this research. For example, in lines 155,156 about the TCP and UDP flood attacks, the authors should provide more evidence to support the statement.

There is no issue with the structure of the article. Some formats in this paper may not be standard, but I can follow the paper with sufficient clarity as a reader. Figures and tables are also written in a good format.

The paper suggests 4 research questions, and the results are comprehensive and sufficiently self-contained. The report includes all results coherently relevant to the research questions. Formal results have already included clear definitions of all terms and theorems, detailed proofs, and measurements.

Experimental design

I believe that this paper is original, and the primary research fits within the aim and scope of the PeerJ Computer Science journal.

The research question is sufficiently well defined, relevant & meaningful. The research fills an identified knowledge gap.
The submission should clearly define the research question, which must be relevant and meaningful. The knowledge gap being investigated should be identified, and statements about how the study contributes to filling that gap.

The investigation has been conducted rigorously, and it conforms to a high technical standard. I would also state that the research is conducted according to ethical standards, particularly in the IoT Security Space.

Validity of the findings

Impact and novelty are sufficiently assessed. The methods described do not provide sufficient information to be reproducible by another investigator. Unless the authors provide the complete programmable codes, then other researchers could replicate these findings.
All underlying data are sufficiently robust, statistically sound, and well-controlled.

Conclusions are sufficiently well stated and linked to the original research question. I would like to comment on lines 93-102; in conclusion, I am not convinced whether this aim has already been achieved. Should this aim is successfully achieved, how would it substantially benefit humanity.
In general, the conclusions are appropriately stated, connected to the original question investigated, and supported the results with clarity. A well-controlled experimental intervention sufficiently supports claims of a causative relationship.

Reviewer 3 ·

Basic reporting

The authors presented their work entitled, Detecting Security Attacks in 2 Cyber-Physical Systems: A Comparison of 3 Mule and WSO2 Intelligent IoT Architectures. Where authors claims that The Internet of Things (IoT) paradigm keeps growing, and many different IoT devices, such as smartphones and smart appliances, are extensively used in smart industries and smart cities. The benefits
of this paradigm are obvious, but these IoT environments have brought with them new challenges,
such as detecting and combating cybersecurity attacks against cyber-physical systems. This paper
addresses the real-time detection of security attacks in these IoT systems through the combined used
of Machine Learning (ML) techniques and Complex Event Processing (CEP). In this regard, in the past
we proposed an intelligent architecture that integrates ML with CEP, and which permits the definition of
event patterns for the real-time detection of not only specific IoT security attacks, but also novel attacks
that have not previously been defined. Our current concern, and the main objective of this paper, is
to ensure that the architecture is not necessarily linked to specific vendor technologies and that it can
be implemented with other vendor technologies while maintaining its correct functionality. We also set
out to evaluate and compare the performance and benefits of alternative implementations. This is why
the proposed architecture has been implemented by using technologies from different vendors: firstly,
the Mule Enterprise Service Bus (ESB) together with the Esper CEP engine; and secondly, the WSO2
ESB with the Siddhi CEP engine. Both implementations have been tested in terms of performance and
stress, and they are compared and discussed in this paper. The results obtained demonstrate that both
implementations are suitable and effective.
However, I can see great room for improvement in multiple sections.
1. The abstract does not align with the paper actual findings and conclusion as well.
2. Figures 2, 3 and a few more are not readable clearly
3. Too many figures, while elaboration is not extensive, authors may elaborate properly.
4. Conclusion requires revision, it can be concise with the findings only, currently, it is too lengthy.
5. Proofread is highly recommended.
6. May elaborate more on their proposed approach.
7. Provided references are better enough. However, a few of them are missing information, and can be strengthen further.

Experimental design

The authors presented their work entitled, Detecting Security Attacks in 2 Cyber-Physical Systems: A Comparison of 3 Mule and WSO2 Intelligent IoT Architectures. Where authors claims that The Internet of Things (IoT) paradigm keeps growing, and many different IoT devices, such as smartphones and smart appliances, are extensively used in smart industries and smart cities. The benefits
of this paradigm are obvious, but these IoT environments have brought with them new challenges,
such as detecting and combating cybersecurity attacks against cyber-physical systems. This paper
addresses the real-time detection of security attacks in these IoT systems through the combined used
of Machine Learning (ML) techniques and Complex Event Processing (CEP). In this regard, in the past
we proposed an intelligent architecture that integrates ML with CEP, and which permits the definition of
event patterns for the real-time detection of not only specific IoT security attacks, but also novel attacks
that have not previously been defined. Our current concern, and the main objective of this paper, is
to ensure that the architecture is not necessarily linked to specific vendor technologies and that it can
be implemented with other vendor technologies while maintaining its correct functionality. We also set
out to evaluate and compare the performance and benefits of alternative implementations. This is why
the proposed architecture has been implemented by using technologies from different vendors: firstly,
the Mule Enterprise Service Bus (ESB) together with the Esper CEP engine; and secondly, the WSO2
ESB with the Siddhi CEP engine. Both implementations have been tested in terms of performance and
stress, and they are compared and discussed in this paper. The results obtained demonstrate that both
implementations are suitable and effective.
However, I can see great room for improvement in multiple sections.
1. The abstract does not align with the paper actual findings and conclusion as well.
2. Figures 2, 3 and a few more are not readable clearly
3. Too many figures, while elaboration is not extensive, authors may elaborate properly.
4. Conclusion requires revision, it can be concise with the findings only, currently, it is too lengthy.
5. Proofread is highly recommended.
6. May elaborate more on their proposed approach.
7. Provided references are better enough. However, a few of them are missing information, and can be strengthen further.

Validity of the findings

The authors presented their work entitled, Detecting Security Attacks in 2 Cyber-Physical Systems: A Comparison of 3 Mule and WSO2 Intelligent IoT Architectures. Where authors claims that The Internet of Things (IoT) paradigm keeps growing, and many different IoT devices, such as smartphones and smart appliances, are extensively used in smart industries and smart cities. The benefits
of this paradigm are obvious, but these IoT environments have brought with them new challenges,
such as detecting and combating cybersecurity attacks against cyber-physical systems. This paper
addresses the real-time detection of security attacks in these IoT systems through the combined used
of Machine Learning (ML) techniques and Complex Event Processing (CEP). In this regard, in the past
we proposed an intelligent architecture that integrates ML with CEP, and which permits the definition of
event patterns for the real-time detection of not only specific IoT security attacks, but also novel attacks
that have not previously been defined. Our current concern, and the main objective of this paper, is
to ensure that the architecture is not necessarily linked to specific vendor technologies and that it can
be implemented with other vendor technologies while maintaining its correct functionality. We also set
out to evaluate and compare the performance and benefits of alternative implementations. This is why
the proposed architecture has been implemented by using technologies from different vendors: firstly,
the Mule Enterprise Service Bus (ESB) together with the Esper CEP engine; and secondly, the WSO2
ESB with the Siddhi CEP engine. Both implementations have been tested in terms of performance and
stress, and they are compared and discussed in this paper. The results obtained demonstrate that both
implementations are suitable and effective.
However, I can see great room for improvement in multiple sections.
1. The abstract does not align with the paper actual findings and conclusion as well.
2. Figures 2, 3 and a few more are not readable clearly
3. Too many figures, while elaboration is not extensive, authors may elaborate properly.
4. Conclusion requires revision, it can be concise with the findings only, currently, it is too lengthy.
5. Proofread is highly recommended.
6. May elaborate more on their proposed approach.
7. Provided references are better enough. However, a few of them are missing information, and can be strengthen further.

Additional comments

The authors presented their work entitled, Detecting Security Attacks in 2 Cyber-Physical Systems: A Comparison of 3 Mule and WSO2 Intelligent IoT Architectures. Where authors claims that The Internet of Things (IoT) paradigm keeps growing, and many different IoT devices, such as smartphones and smart appliances, are extensively used in smart industries and smart cities. The benefits
of this paradigm are obvious, but these IoT environments have brought with them new challenges,
such as detecting and combating cybersecurity attacks against cyber-physical systems. This paper
addresses the real-time detection of security attacks in these IoT systems through the combined used
of Machine Learning (ML) techniques and Complex Event Processing (CEP). In this regard, in the past
we proposed an intelligent architecture that integrates ML with CEP, and which permits the definition of
event patterns for the real-time detection of not only specific IoT security attacks, but also novel attacks
that have not previously been defined. Our current concern, and the main objective of this paper, is
to ensure that the architecture is not necessarily linked to specific vendor technologies and that it can
be implemented with other vendor technologies while maintaining its correct functionality. We also set
out to evaluate and compare the performance and benefits of alternative implementations. This is why
the proposed architecture has been implemented by using technologies from different vendors: firstly,
the Mule Enterprise Service Bus (ESB) together with the Esper CEP engine; and secondly, the WSO2
ESB with the Siddhi CEP engine. Both implementations have been tested in terms of performance and
stress, and they are compared and discussed in this paper. The results obtained demonstrate that both
implementations are suitable and effective.
However, I can see great room for improvement in multiple sections.
1. The abstract does not align with the paper actual findings and conclusion as well.
2. Figures 2, 3 and a few more are not readable clearly
3. Too many figures, while elaboration is not extensive, authors may elaborate properly.
4. Conclusion requires revision, it can be concise with the findings only, currently, it is too lengthy.
5. Proofread is highly recommended.
6. May elaborate more on their proposed approach.
7. Provided references are better enough. However, a few of them are missing information, and can be strengthen further.

---

## Round 0.2 · accepted · Accept

The paper can be Accepted with no further revisions.

·

Basic reporting

All the changes have been done as per suggestions of reviewers. Current form of paper is suitable for publication.

Experimental design

Fine

Validity of the findings

Fine

Additional comments

NA

·

Basic reporting

The authors wrote the article in good professional English. As the reader, I can easily follow the storyline with ease and clarity. There is no issue with the structure of the article. Some formats in this paper may not be standard, but I can follow the paper with sufficient clarity as a reader. Figures and tables are also written in a good format.

Experimental design

The paper contains relevant information adequate to justify publication, demonstrating a proper understanding of the relevant literature. No considerable work is ignored. The paper's argument is already built on an appropriate base of theory, concepts, and other ideas. The research or equivalent intellectual work on which the article is already based has been well designed. And the methods have been employed appropriately.

Validity of the findings

The results are presented clearly and analyzed appropriately, and the conclusions adequately tie together the other elements of the paper. The paper already identifies any implications practice and the implications consistent with the findings and conclusions of the article.  The article clearly expresses its case, measured against the technical language of the field.

I would be happy if this paper could be published in this journal. A reasonable effort has been made. By reviewing this paper, I gained more insight into IoT, an essential application for blockchain technology that I am working on.

Additional comments

To begin with, I would like to thank the authors for considering the Peer-J Computer Science as a publication outlet for their paper "Detecting Security Attacks in 2 Cyber-Physical Systems: A Comparison of 3 Mule and WSO2 Intelligent IoT Architectures". The report addresses the real-time detection of security attacks in these IoT systems through the combined use of Machine Learning (ML) techniques and Complex Event Processing (CEP).

The paper has improved in comparison to its previous version.

Overall this paper is a clear, concise, and well-written manuscript. The introduction is relevant and theory-based. Sufficient information about the previous study findings is presented for readers to follow the present study rationale and procedures. The methods are generally appropriate. Good luck.

Reviewer 3 ·

Basic reporting

The authors address all the comments and concerns. The paper stands for acceptance now.

Experimental design

The authors address all the comments and concerns. The paper stands for acceptance now.

Validity of the findings

The authors address all the comments and concerns. The paper stands for acceptance now.

Additional comments

The authors address all the comments and concerns. The paper stands for acceptance now.